# Impact of evolving greenhouse gas forcing on the warming signal in regional climate model experiments

S. Jerez[1,2], J.M. López-Romero[1], M. Turco [3], P. Jiménez-Guerrero [1], R. Vautard [4] & J.P. Montávez [1]

Variations in the atmospheric concentrations of greenhouse gases (GHG) may not be included as external forcing when running regional climate models (RCMs); at least, this is a non-regulated, non-documented practice. Here we investigate the so far unexplored impact of considering the rising evolution of the $CO_2$, $CH_4$, and $N_2O$ atmospheric concentrations on near-surface air temperature (TAS) trends, for both the recent past and the near future, as simulated by a state-of-the-art RCM over Europe. The results show that the TAS trends are significantly affected by 1–2 K century$^{-1}$, which under 1.5 °C global warming translates into a non-negligible impact of up to 1 K in the regional projections of TAS, similarly affecting projections for maximum and minimum temperatures. In some cases, these differences involve a doubling signal, laying further claim to careful reconsideration of the RCM setups with regard to the inclusion of GHG concentrations as an evolving external forcing which, for the sake of research reproducibility and reliability, should be clearly documented in the literature.

[1] Regional Atmospheric Modeling Group, Department of Physics, University of Murcia, 30100 Murcia, Spain. [2] Laboratório Associado IDL, Faculdade de Ciencias, Universidade de Lisboa, 1749-016 Lisboa, Portugal. [3] Department of Applied Physics, University of Barcelona, 08028 Barcelona, Spain. [4] Laboratoire des Sciences du Climat et de l'Environnement (LSCE), IPSL, CEA-CNRS-UVSQ, 91191 Gif sur Yvette, France. Correspondence and requests for materials should be addressed to S.J. (email: sonia.jerez@gmail.com)

Global warming manifests itself and impacts heterogeneously across the globe[1–4]. The translation of global temperature increases and targets, such as the encouraged long-term goal of keeping the increase in global mean temperature to well below 2 °C above pre-industrial levels[5], into regional impacts and vulnerabilities is of major importance in order to raise awareness of societies and decision-makers[6]. In order to better describe the consequences of climate change at a proper spatial scale, dynamical downscaling techniques are typically applied to elucidate global warming fingerprints at regional scales and to produce policy-relevant information tailored to the needs of the potentially affected socioeconomic sectors (e.g., energy, food, and health). The dynamical downscaling approach is based on the use of Regional Climate Models (RCMs) running over limited geographical domains with boundary conditions given by Global Circulation Models (GCMs) to be downscaled over the finer RCM spatial grid; although there is another dynamical downscaling approach given by variable resolution GCMs. During the past decades, RCMs have been commonly used by a wide community of researchers and under the umbrella and incentive of many international initiatives, such as CORDEX[7,8], sponsored by the World Climate Research Programme (WCRP), whose aim is to advance and coordinate the science and application of regional climate downscaling through global partnerships.

The main added value of high-resolution RCMs is the development of the fine scale, which reaches its optimum by considering large regional domains[9–11]. Although the RCMs explicitly solve mesoscale atmospheric processes, there are still small-scale processes that need to be parameterized[12]. In particular, radiation schemes are used to manage the radiative forcing, which is affected by the concentration of greenhouse gases (GHG) in the atmosphere. Some radiation schemes, as implemented in RCMs, prescribe GHG concentrations to fixed values by default (probably because they were primarily conceived for weather forecast applications or short-term runs). For instance, in the widely used Weather Research and Forecasting (WRF) model[13], the inclusion of variable GHG concentrations as an evolving external forcing is an available option only from its 3.5 version (launched in April 2013) and is not used in the default compilation. The use of constant GHG concentrations in long-term RCM runs leads to the implicit assumption that change in the GHG atmospheric composition has little effect on the regional climate, or that its effect is sufficiently included through the GCMs-provided boundary conditions. No theoretical or empirical scientific evidence supports, however, either of these assumptions. The latter contrasts in particular with the recommended use of large domains as this results, in turn, in weak control by the lateral boundary conditions[9].

Blind to this issue, the benchmark CORDEX framework, for instance, provides scant information on the RCMs implementations and capabilities of using the same climate external forcing data (i.e., GHG atmospheric concentrations) as the parent GCMs, in spite of its triggering factor in coordinating experimental setups among modeling groups worldwide. In fact, while most of the RCMs contributing to the European branch of CORDEX (EURO-CORDEX[14,15]) do actually include varying GHG concentrations in their runs (e.g., RACMO, RCA, ALADIN, CCLM, and REMO), others, as far as we are aware to date, do not (in particular some WRF and HIRHAM runs). This information had to be gathered by personal communication with the modeling groups, since no documentation at this regard is publicly available. In contrast to the well documented experimental design (including the GHG forcing) within GCM coordination frameworks such as the Climate Model Intercomparison Project (CMIP)[16,17], this aspect of the RCMs setup is undocumented in the current scientific literature (not only within the EURO-CORDEX framework, but also anywhere else aside from

very few exceptions[18]), which, as little, goes against the code of good practice that should guarantee reproducible research[19]. Moreover, while many works deal with the sensitivity of RCM runs, for example, to the domain spatial resolution and the choice of parameterization schemes, the inclusion or not of the forcing due to the variations in the atmospheric concentrations of GHG has so far received very little attention (find a single attempt in ref.[20]). In spite of its likely paramount importance[8,21] and practical relevance for the accuracy of regional climate projections, the sensitivity of RCM runs to evolving GHG concentrations has not yet been properly established.

The impact of varying GHG concentrations in RCM runs should be most apparent with regard to temperature trends. In this sense, several studies[22–26] have already warned about the poor ability of RCMs to reproduce the magnitude of observed near-surface temperature (TAS) trends along the recent past, reporting an overall systematic underestimation that even worsen the signal from the GCMs driving data. As possible causes, they pointed out inadequacies in the characterization and modeling of basic processes and features, such as cloud formation, aerosol variations, and soil properties, which are certainly what climate models miss most[27]. However, the focus so far has not been on the likely relationship of this issue to how GHG concentrations are treated in RCM runs. The importance of the matter, with its immediate implications for the regional climate modeling community, motivated us to shed light on it.

Therefore, here we investigate the sensitivity of TAS trends to year-to-year varying GHG concentrations in RCM simulations by analyzing pairs of historical (1951–2005) and scenario (2006–2050) regional climate simulations run with WRF over Europe. Each counterpart was performed allowing or not allowing the GHG concentrations to vary in the RCM (VAR and CTE experiments, respectively: VAR from varying GHG concentration values; CTE from constant values, i.e., those corresponding to the year 2005 in our experimental setup). In order to obtain the highest sensitivity to the variations of the GHG atmospheric concentrations, in the scenario period we considered the Representative Concentration Pathway 8.5 (RCP8.5)[28,29], the most challenging one with the highest increases of GHG emissions among those considered in the latest IPCC report[1], although in the first half of the 21st century it is very similar to the more moderate RCP4.5. The WRF setup and the GCM data driving WRF (retrieved from the r1i1p1 MPI-ESM-LR CMIP5-experiment[30,31]) are described in detail in Methods. For the historical period, WRF was additionally driven by the ERA20C reanalysis data[32,33] (hereafter referred to as ERA). These reanalysis-driven experiments allowed us to elucidate the contribution of varying GHG concentrations also in RCM hindcast runs.

The fingerprint of evolving GHG concentrations in the regional warming is apparent and robust in all cases, under both past and future conditions, involving even doubled signals under 1.5 °C global warming in some areas (eastern Europe) and seasons (winter). Such a non-negligible impact calls for regulating efforts to include and document systematically the time-varying GHG forcing in the RCM runs as a good practice.

## Results

**Temperature trends**. Figure 1 shows the linear trends of the yearly-mean TAS series retrieved from the global datasets and the regional experiments (WRF CTRL configuration, see Table 1): ERA and ERA-driven WRF CTE and VAR experiments in the period 1951–2005, GCM and GCM-driven WRF CTE and VAR experiments in the period 1951–2005, and GCM and GCM-driven WRF CTE and VAR experiments in the period 2006–2050. The differences between the VAR and CTE experiments and

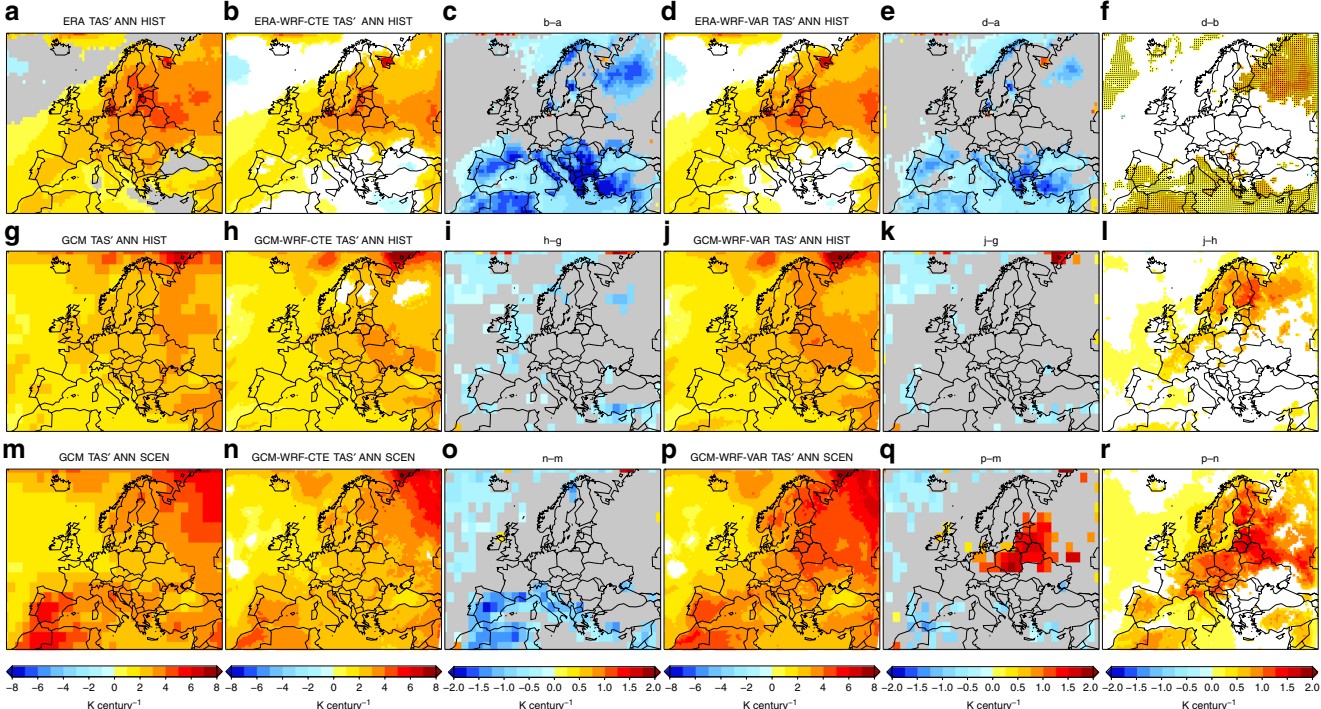

**Fig. 1** Near-surface air temperature trends from global datasets and regional simulations. Trends of the simulated yearly-mean time series of TAS in the historical period 1951–2005 (first and second rows) and the scenario period 2006–2050 (third row). The first row depicts the results from ERA (**a**) and ERA-driven WRF experiments (CTRL configuration, see Table 1), CTE (**b**) and VAR (**d**), along with the differences between ones and others: CTE minus ERA (**c**), VAR minus ERA (**e**), and VAR minus CTE (**f**). Similarly, the second and third rows (**g–r**) depict the results from GCM and GCM-driven WRF experiments. Only significant values ($p < 0.1$) are shown. The points in the last column indicate that the magnitude of the difference between the VAR and CTE experiments is equal to or greater than the magnitude of the signal from the respective CTE experiment. Units: K century$^{-1}$

| | Radiative scheme | Microphysics scheme | Cumulus scheme |
|---|---|---|---|
| Exp. 1 | CAM shortwave and longwave schemes[48] | Lin et al. scheme[45] | Grell 3D ensemble scheme[43,44] |
| Exp. 2 (CTRL) | RRTMG shortwave and longwave schemes[42] | Lin et al. scheme[45] | Grell 3D ensemble scheme[43,44] |
| Exp. 3 | CAM shortwave and longwave schemes[48] | Morrison 2—moment scheme[49] | Kain-Fritsch scheme[50] |
| Exp. 4 | RRTMG shortwave and longwave schemes[42] | Morrison 2—moment scheme[49] | Kain-Fritsch scheme[50] |

**Table 1 Radiative, microphysics, and cumulus schemes used in the multi-physics ensemble of WRF simulations. We denote as CONTROL (CTRL) configuration the one corresponding to Exp. 2**

between these and the global datasets are also provided. The Supplementary Material (Supplementary Figs 1–9) provides the same analysis for the DJF and JJA-mean TAS series and for maximum (Tx) and minimum (Tn) temperatures.

All the global and regional datasets and experiments usually depict positive trends, which are more marked over land than sea. The spatial patterns are very similar for TAS, Tx and Tn, with the largest signals (up to 6 K century$^{-1}$ at the annual scale) appearing over central and northern Europe for the recent past, and also along the south-western Mediterranean strip for the future. By seasons, the signals increase to 8 K century$^{-1}$ in winter (north-eastern areas, past period) and summer (south-western areas, future period).

Both WRF experiments (VAR and CTE) reproduce the most outstanding features of the spatial patterns retrieved from their respective global driving datasets, while showing some important discrepancies. For instance, at the annual scale, both ERA-driven WRF experiments provide weaker TAS trends in Mediterranean Europe and north-eastern areas compared to the results from ERA (Fig. 1, first row). This is more marked in the CTE experiment than in the VAR, the differences between them being statistically significant over both such areas, with higher values

from the VAR experiment, up to 1 K century$^{-1}$ higher (which involves doubling the signal), than from the CTE one (Fig. 1f). Note, however, that the differences between the VAR and CTE experiments in reproducing TAS trends are smaller than between each WRF realization and ERA, thus the underestimation of TAS trends still persists in the ERA-driven VAR simulation. Generally, the differences between the CTE and VAR runs are larger in the GCM-driven experiments (Fig. 1l, r), particularly in the future period. In that case (Fig. 1r), statistically significant differences ranging from 0.5 to 1.5 K century$^{-1}$ appear over most of Europe, widely reaching 2 K century$^{-1}$ over the central and north-eastern regions in the DJF and JJA seasonal analysis (Supplementary Figs 2, 3), which again involves doubling the signal in these areas. Very similar results are found for Tx and Tn (Supplementary Figs 4–9), which support the structural character of the differences between the VAR and CTE approaches reported. The fact that the differences between the VAR and CTE experiments are notably larger over land than over the sea merely reflects the key control that the sea surface temperature (SST) exerts on TAS, as SST remains identical in the VAR and CTE regional simulations. This also could reflect the influence of

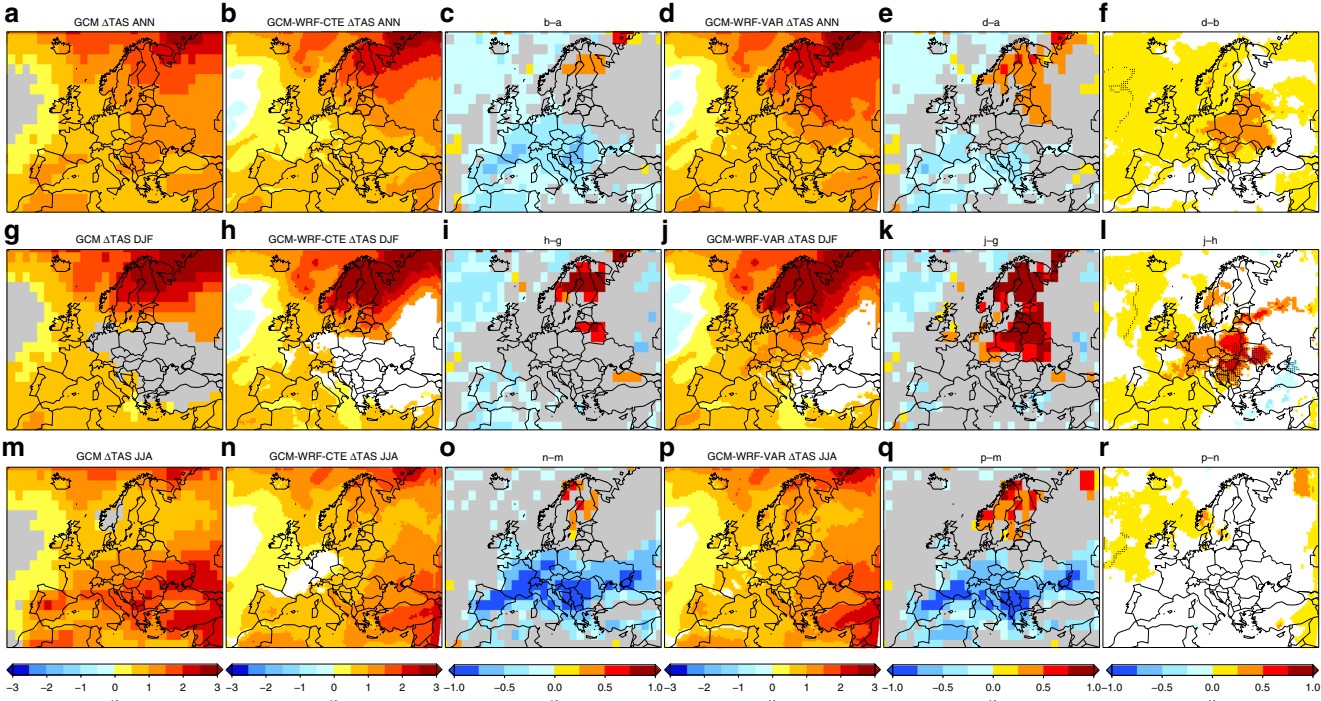

**Fig. 2** Near-surface air temperature projections under 1.5 °C global warming from global and regional simulations. Projected changes (2011–2040 vs. 1971–2000 climatologies) in year-mean (first row), DJF-mean (second row), and JJA-mean (third row) TAS from GCM (first column, panels **a**, **g**, **m**) and GCM-driven WRF experiments (CTRL configuration, see Table 1), CTE and VAR (second and fourth columns, panels **b**, **h**, **n**, and **d**, **j**, **p**, respectively), along with the differences between ones and others: CTE minus GCM (third column, panels **c**, **i**, **o**), VAR minus GCM (fifth column, panels **e**, **k**, **q**), and VAR minus CTE (sixth column, panels **f**, **l**, **r**). The points in the last column indicate that the magnitude of the difference between the VAR and CTE experiments is equal to or greater than the magnitude of the signal from the respective CTE experiment. Only significant values (p < 0.1) are shown. Units: K

the atmospheric composition on the well-known land-atmosphere feedback (higher temperatures leading to drier soils, leading to higher temperatures, and so on[34,35]).

**Projections under 1.5 °C warming**. Next we investigated how the differences in TAS trends between the VAR and CTE experiments translate into different climate change signals. For that, scenario simulations were analyzed with the focus on impacts under 1.5 °C global warming, a limit threshold (agreed in the 2015 Paris climate conference COP21 with the aim of minimizing the risks and impacts associated with the consequences of climate change[5]) that will guide a special report by the IPCC planned for 2018[36]. Thus we computed changes from the VAR and CTE experiments (WRF CTRL configuration, see Table 1) in the period 2011–2040 (when global mean temperature from the used GCM is 1.5 °C warmer than the pre-industrial level) with respect to the reference period 1971–2000 (see Methods).

Figure 2 provides climate change patterns for TAS (at annual and seasonal scales) for the 1.5 °C global warming future period (2011–2040), compared to the historical period (1971–2000), as retrieved from GCM and GCM-driven WRF simulations, both CTE and VAR. Analogous plots (with very similar appearances) for Tx and Tn are provided in the Supplementary Material (Supplementary Figs 10–12). Increases over 2 K, up to 3 K at most, are projected from all the datasets and seasons (annual, winter, and summer scales), with the patterns from the regional experiments resembling the overall structure of the patterns from their driving GCM (Fig. 2, first, second, and fourth columns). The differences between the former and the latter are negative (positive) in southern (north-eastern) areas, generally smaller (larger) for the VAR experiment than for the CTE but for summer, (Fig. 2, third and fifth columns). Hence, the differences between the VAR and CTE experiments, when they are used to

project changes in TAS under 1.5 °C global warming, depict overall positive signals (the largest changes are projected from the VAR experiment) affecting most of continental Europe with significant values around 0.5 K both at the annual scale and for winter (Fig. 2 last column). Although the differences between the VAR and CTE experiments in the TAS projections are generally smaller than the differences between the WRF and GCM simulations, the former implies that the change signal from the VAR realizations doubles the change signal from the CTE ones in some areas of central Europe. This is most evident for Tn, when the differences between the VAR and CTE change patterns spread and become the strongest (up to 1 K, Supplementary Figs 11–12). In turn, this is least evident, negligible indeed, when the regional climate simulations are conducted over small area domains and thus strongly tied by the lateral boundary conditions (Supplementary Figs 13).

**Beyond a particular case study**. The attribution of the differences shown in the last columns of Figs. 1 and 2 to the varying GHG concentrations in the VAR runs is not only straightforward, due to our experimental design, but also consistent with what one would expect. Nevertheless, in order to reinforce our message, here we provide evidence of the coherence and consistency of our results in a wider context. For that we run and analyze the signals from a multi-physics ensemble of regional simulations. It should be noted that different physical configurations of regional models provide the same range of uncertainty (or ensemble spread) as an ensemble of different models, as they may indeed rely on different physical configurations[37–40]. Our ensemble includes modifications in the radiative, microphysics and convective schemes, totalizing a number of four ensemble members (Table 1, see Methods).

Figure 3 summarizes the results from the ensemble as regards the sensitivity of the TAS projections under 1.5 °C global warming to the varying GHG approach within the regional

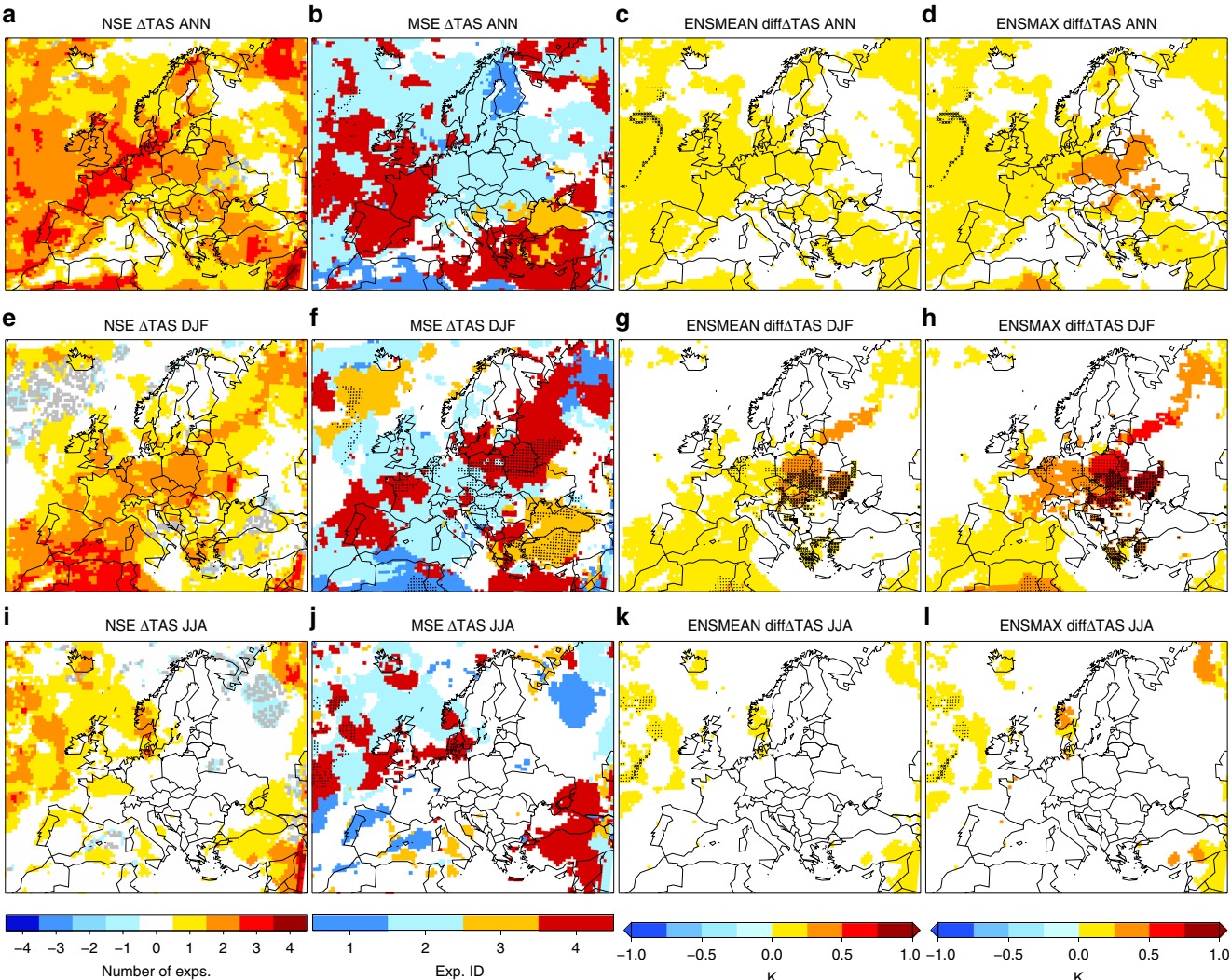

**Fig. 3** Sensitivity of near-surface air temperature projections to the GHG handling in an ensemble of regional simulations. Number of sensitive experiments (NSE, first column, panels **a**, **e**, **i**) and most sensitive experiment (MSE, second column, panels **b**, **f**, **j**, numbers referred to the experiments identifiers of Table 1), within the ensemble of WRF runs (Table 1), to the GHG handling by the RCM when projecting TAS changes under 1.5 °C global warming (ΔTAS, assessed at annual, DJF and JJA scales in first, second and third rows, respectively). The NSE is computed by assigning 1 to those experiments in which the varying GHG approach provides significantly ($p < 0.1$) higher values of ΔTAS than the constant GHG approach, −1 to those in which ΔTAS is significantly lower in the varying GHG counterpart of each experiment, 0 if there is no significant difference between both approaches, and then summing over the four experiments; with the color gray indicating the mix of 1 and −1. MSE denotes the experiment in which the difference in ΔTAS between approaches is highest if it is statistically significant; if not, it is white. The points in the second column indicate that ΔTAS in the MSE is at least double when GHG vary. The third and fourth columns (panels **c**, **g**, **k**, and **d**, **h**, **l**) depict, respectively, the ensemble mean and maximum values of the differences in ΔTAS (units: K) between approaches over the grid points in which at least two experiments provide significant values of such differences; otherwise, it is white. The points, crosses, stars, and squares in the third and fourth columns indicate that ΔTAS is at least double when GHG vary in one, two, three, or four experiments, respectively

model (see also Supplementary Figs 14–19 for detailed plots and the results for *T*x and *T*n). The first column depicts the number of sensitive experiments (NSE) to varying GHG concentrations when projecting TAS changes, with positive/negative values indicating larger/smaller TAS increments in the VAR than in the CTE simulations in those sensitive experiments. Over wide areas, three out of four experiments show significant sensitivity to the varying GHG approach, especially when the TAS projections are assessed at the annual scale, with higher TAS increases in the VAR experiments than in the CTE, overwhelmingly. The second column displays the most sensitive experiment (MSE) among all the ensemble members to the varying GHG approach when projecting TAS changes to the future. While it should be acknowledged that one of the two radiation schemes included in the ensemble (the RRTMG

scheme, Exps. 2 and 4 in Table 1) leads to higher differences in TAS projections between GHG approaches than the other (the CAM scheme, Exps. 1 and 3 in Table 1) (see Supplementary Figs 14–16), the MSE patterns show a quite heterogeneous structure demonstrating that all the configurations are sensitive to the varying GHG approach. In fact, all of them provide signals of TAS change that, in some regions, in the VAR experiment are double than those in the CTE one (see points in second column and Supplementary Figs 14–19). The third and fourth columns depict ensemble mean and maximum values of the differences between the VAR and CTE experiments in the TAS projections. The patterns are similar to those of the last column of Fig. 2, thus further proving and confirming the coherence of our results under an ensemble of climate change simulations.

## Discussion

Our analysis yields statistically significant and noticeable differences in the TAS trends between regional climate simulations using the two different assumptions, constant or varying GHG concentrations in the RCM, when conducted for both the recent past (including the case of reanalysis-driven simulations) and the near future. Other variables and/or RCMs may show different sensitivity, which could be worth to investigate. Anyhow, the worrying issue is that whether the forcing due to the variations in the atmospheric concentrations of GHG is or not included in the regional climate model runs is not sufficiently documented in the scientific literature. This lack of information, together with (1) the fact that the default compilation in one of the most used (and most recent) RCMs, the WRF model, does not include such time-varying forcing, and (2) the information that these authors have gathered regarding the non-regulated practice in this regard within the outstanding EURO-CORDEX initiative, alarmingly incites doubt to emerge.

Here we intend to appeal to the regional climate modeling community worldwide by identifying the need to clarify and document the approach when reporting regional climate information. It is clear that the discrepancies between both approaches are large enough for better information on the setup of RCMs to be required, especially when the regional climate simulations are conducted over wide domains and contribute to initiatives demanding coordination efforts (e.g. CORDEX, IPCC-related simulations). Special attention should be paid to the design of the regional climate modeling experiments that may involve the use of data by a wide community of potential users, including the impacts community.

## Methods

**Experiments**. The 3.6.1 version of the WRF model[13] was used to run the regional climate simulations over Europe with a spatial resolution of 0.44° in latitude and longitude, using the EURO-CORDEX compliant domain[14]. For the historical period (1951–2005), both the ERA20C reanalysis[32,33], which assimilates observations of surface pressure and surface marine winds and has an horizontal resolution of approximately 125 km, and the GCM CMIP5-experiment[16,17] r1i1p1 MPI-ESM-LR historical run[30] with an approximate horizontal resolution of 200 km over land (hereafter referred to as ERA and GCM data, respectively) were used to initialize and drive (without nudging) the WRF model. Both global datasets incorporate the year-to-year varying annual concentrations of the $CO_2$, $CH_4$, and $N_2O$ GHG (as estimated from inventory data[28]), which show a continuous positive trend[1]. For the scenario period (2006–2050), the GCM CMIP5-experiment RCP8.5-forced r1i1p1 MPI-ESM-LR run[31] was used. RCP8.5 depicts the highest radiative forcing along the XXI century among all RCPs[28,29], with doubled $CO_2$, $CH_4$, and $N_2O$ concentrations (these are again annual estimations) by 2050 compared to the last record of the historical period.

The first setup of the WRF model (CTRL configuration, see Table 1) included the Noah land surface model[41], RRTMG shortwave and longwave schemes[42], Grell 3D ensemble scheme for cumulus[43,44], Lin scheme for microphysics[45] and Yonsei University scheme for the boundary layer[46]. To construct the multi-physics ensemble we used all the WRF configurations listed in Table 1. The choice includes the only two radiative schemes handling variable GHG concentrations in WRF, the two most different (single-moment and two-moment) microphysics parameterizations and, linked with that (not all microphysics schemes are compatible with all cumulus schemes), one of the simplest and one of the most complex convective schemes. Based on previous results[40], these configurations yield results that cover a wide range of the model spread. Simulations other than the CTRL one span only the periods 1971–2000 and 2011–2040, which are enough to assess impacts under 1.5 °C global warming. Also, the GCM-driven CTRL experiments were replicated for these periods over a smaller area domain, with latitude ranging from 44.8° to 53.5° and longitude from 16.2° to 30° (see Supplementary Fig. 13).

In practice, the simulated periods were split into 5-year periods that were then continuously run with a spinup period of 4 months (time-slice approach). Updates of the boundary conditions, from either ERA or GCM, were performed every 6 h at the borders of the regional domain. The outputs were recorded at one-hour intervals.

Each WRF realization was run twice: considering static concentrations of $CO_2$, $CH_4$, and $N_2O$ GHG, fixed to their values in 2005 (the by-default option in WRF; CTE experiments), and considering their evolution, their annual estimated values, along the simulated periods as it was included in the driving global datasets (VAR experiments).

**Data analysis**. The analysis first deals with the temporal trends of TAS over both historical and scenario periods: differences between the VAR and CTE experiments, and between these and their driving global datasets (ERA and GCM). The temporal trends were computed as the linear trends of the yearly-mean series of TAS at either annual or seasonal scales (winter: December to February, DJF, averages; summer: June to August, JJA, averages). The daily maximum and minimum TAS ($Tx$ and $Tn$, respectively) series were also yearly or seasonally averaged and included in the analysis. The linear trends were computed using the Theil–Sen estimator, a non-parametric method that defines the slope as the median among all possible lines through pairs of points. This approach is less sensitive to outliers than the classic least-squares method and is more accurate in case of non-normal data. Trend significance was estimated applying the standard Mann–Kendall trend test. Statistical significance of trend differences (i.e., between datasets) was equally estimated but using the difference time series (i.e., series of TAS, $Tx$ or $Tn$ differences between datasets). This method reduces noise levels by subtracting the variability common to both datasets being faced[47]. In all cases, we imposed the threshold $p < 0.1$.

Second, the analysis deals with climate change regional impacts on TAS, $Tx$ and $Tn$, as depicted by the VAR and CTE experiments under 1.5 °C global warming above pre-industrial levels. For that, following the common procedure[6], change patterns were constructed by subtracting the averaged values of TAS, $Tx$ and $Tn$ over the 30-year future period centered in the year Y when global mean temperature rises 1.5 °C higher than in the pre-industrial era, and their averages over the recent past period 1971–2000. To define Y, we considered the 30-year running mean series of global mean TAS retrieved from the RCP8.5-forced GCM simulation used to feed WRF and looked for an increment of 1.04 °C compared to the record corresponding to the average of the period 1971–2000. Until this base period, 1971–2000, the pre-industrial warming (since the record corresponding to the average of the period 1881–1910) had been 0.46 °C according to the observations[6]; that is why we looked for an increment of 1.04 °C since 1971–2000. Applying this methodology, Y was found to be 2025. Thus the future period considered here to construct the change patterns of TAS, $Tx$ and $Tn$ was 2011–2040. Statistical significance of the signals appearing in the change patterns and the patterns of differences (i.e., between datasets) was accounted for by applying a two-tailed $t$-test. Again, we imposed the threshold $p < 0.1$.

To compare the signals retrieved from the WRF experiments and from their driving global datasets at the grid point level, the former were bilinearly spatially interpolated to the respective global grids.

**Data availability**. All datasets used in this study are publicly available. GCM and ERA data can be retrieved from the websites of the Climate and Environmental Retrieval and Archive (https://www.dkrz.de/daten-en/cera) and the European Center for Medium Range Weather Forecast (https://www.ecmwf.int/en/forecasts/datasets/reanalysis-datasets/era-20c, respectively. On behalf of reproducibility, the WRF runs used in this work are available for research purposes by contacting the corresponding author. In any case, the open source WRF code is freely available from https://www.mmm.ucar.edu/weather-research-and-forecasting-model (we used the version 3.6.1), and the WRF namelists as used here are provided in the Supplementary Tables 1 and 2, thus guaranteeing full reproducibility of our experiments. The codes used for the data processing and analysis are based on open source software: the Climate Data Operators (CDO version 1.6.3, functions: remapbil, daymean, daymax, daymin, yearmean, seasmean, and diff) available from https://code.mpimet.mpg.de/projects/cdo/, and the R Project for Statistical Computing (R version 3.2.1, functions: t.test and mannKen, the latter from the R package WQ version 0.4.7) available from https://www.r-project.org/. All codes are as well available from the corresponding author upon request.

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

## Acknowledgements

This study was funded by the Spanish government and the Fondo Europeo de Desarrollo Regional (FEDER) through the project REPAIR (CGL2014-59677-R). S.J. acknowledges the Portuguese Foundation for Science and Technology (FCT) for its support through the FCT Investigator Programme 2015 (IF/01142/2015). M.T. was supported by the Spanish *Juan de la Cierva* Programme (IJCI-2015-26953). Special thanks to Samuel Somot for the helpful discussions on this study.

## Author contributions

S.J. and J.M.L.-R. contributed equally to this paper. S.J., J.M.L.-R., and J.P.M. conceived the original idea of this work, and designed and performed the experiments. S.J., J.M.L.-R., and M.T. performed the analysis and wrote the paper. J.P.M., P.J.-G., and R.V. contributed to the discussion of the results and to writing the paper.

## Additional information

**Competing interests:** The authors declare no competing interests.

