## [Peer Review File · Nature Communications]

Reviewers' comments:

Reviewer #1 (Remarks to the Author):

The authors of the paper in review highlight the poor level of documentation in the current literature of the capability of Regional Climate Models (RCM) to use same climate global forcing data as used by the boundary forcing GCM.

In the framework of the CMIP5 experiment, the climate forcing data list has been agreed upon and the list of recommended and required datasets can be found on the project website (<http://cmip-pcmdi.llnl.gov/cmip5/forcing.html>). The experiment design document (Taylor 2009/2011) describes for the Historical experiment to include:

- atmospheric composition (including CO₂), due to both anthropogenic and volcanic influences
- solar forcing
- emissions or concentrations of short-lived species and natural and anthropogenic aerosols or their precursors.
- land use

while for each of the RCP experiment the prescribed GHG concentration changes were required to facilitate direct comparison between models with and without a carbon cycle component.

For the CORDEX experiment, Gutowsky et al paper of 2016 (WCRP COordinated Regional Downscaling EXperiment WCRP COordinated Regional Downscaling EXperiment (CORDEX): a diagnostic MIP for CMIP6) outlines the importance of GHG concentration as a forcing also for the RCM physical downscaling, together with the land-use and aerosol forcing effects. In Jones 2011 (Jones C, F. Giorgi and G. Asrar, 2011: The Coordinated Regional Downscaling Experiment: CORDEX, An international downscaling link to CMIP5: CLIVAR Exchanges, No. 56, Vol 16, No.2 pages 34-40) it is written:

"The main sources of uncertainty in regional climate change projections can be summarized as: (i) GHG emission uncertainty, (ii) CGCM differences, (iii) CGCM internal variability, (iv) RCD method/model/resolution differences and (v) RCD internal variability. Furthermore, it is also possible that systematic errors in either or both CGCMs and RCMs can affect the realism of simulated climate projections. The relative contribution of all these sources to the total uncertainty will be dependent on the region of interest."

Thus in the CORDEX experiment framework the importance of the GHG forcing also at regional scale is emphasized.

Nevertheless, in net contrast with the high level of detail provided for CMIP5 experiment framework, the CORDEX experiment framework provides scant information on the RCM model capabilities to use the same climate forcing data used by the parent GCMs, and concentrate mostly in describing regional domains and nesting GCM. On the CORDEX project website, in the experiment protocol there is no explicit mention to the climate forcing capabilities of the RCM, and in the data format and archive design no information is required on climate forcing. Only information required is the detail of the boundary GCM forcing.

The authors of the paper in review make two strong claims:

1) on line 51-55 : "Some radiation schemes, as implemented in RCMs, prescribe by default GHG

concentrations to fixed values, assuming that the GHG atmospheric composition change has little effects on the regional climate with the effect of global climate change being sufficiently included through the GCMs-provided boundary conditions."

2) on line 58-60: "In personal communication with various modeling groups involved in the EURO-CORDEX initiative, we found out that some of them include varying GHG concentrations in their RCM runs; while others do not"

The first claim is both in contrast with lines 50-51 of the paper in review "radiative schemes [...] are heavily affected by the concentration of GHG on the atmosphere", the CORDEX experiment aim as described in the Jones and Gutowski papers, and the evidence at large of the climate change as having an anthropogenic source from the GHG concentration increase. This claim is moreover shown to be false by the paper itself. Where this assumption comes from? To my knowledge, no research paper making this claim has shown evidence of it to the scientific community.

The second claim, that some EURO-CORDEX model do not implement varying GHG as prescribed in the RCP used by the parent driving GCM, it must be detailed and a list of this models should be provided. The statement that "[...] we intend to present no judgment on the methodology followed by regional modeling groups worldwide [...]" is not enough on scientific point of view: science should not be interested in not hurting prides, but in spotting errors and correct them to better describe reality.

The authors should at least provide for the EURO-CORDEX experiment they claim to have information about a table of the models with flag if they implement or not varying GHG.

The paper provides a good experiment to outline the importance of using the prescribed GHG concentration data from the RCP in the RCM consistent with the one used by the driving GCM.

On the data analysis, the regional impact of the varying GHG is well detailed and consistent with one would expect but the attribution to model internal variability should be excluded also for completeness by performing a small perturbation experiment of the CTE run.

On the issue of reproducibility, a stronger and more effective request to the community of Regional Climate modelers would be to include any relevant parameter required to run the scenario simulation contributed to the ESGF repository as metadata in the model output file itself. In the authors paper itself, some 10 lines were required to describe the RCM model setup, but for other RCM models where any physical parametrization is tunable, not providing the full list of physically relevant parameters most often prevent any other research group from replicating the experiment and thus falsification may be impossible without direct interaction with the data provider.

Reviewer #2 (Remarks to the Author):

Review of "Towards good practice in regional climate modeling: constant vs. evolving GHG concentrations and its impact on the warming signal" by S. Jerez, J. M. López-Romero, M. Turco, P. Jiménez-Guerrero, R. Vautard and J. P. Montávez

General comments:

The main aim of this paper is to call attention about an often undocumented and certainly understudied practice regarding the inclusion of greenhouse gases (GHG) concentrations as an evolving external forcing within climate simulations using limited-area nested Regional Climate Models. A second aim is to assess the effect of constant Vs. varying GHG by looking at 2-m temperature trends in twin simulations (with/without varying GHG) performed over Europe using a

single RCM driven by two different global datasets (a GCM and a reanalysis). Their results show that temperature recent past trends and near future changes are significantly different when including the effect of GHG within the RCM domain. The general effect is shown to be important when considering annual mean temperatures but also seasonal (JJA and DJF) mean minimum and maximum temperatures.

Since the inception of RCMs nearly 25 years ago, the RCM community has devoted important efforts to assess the sensitivity of a number of experimental choices within the RCM setup including the domain size and position, spatial resolution jump between the RCM and the driving data, nudging technique at the boundaries and in the interior, and others. However, as the authors note in the text, (and as far as I know) this is the first study quantifying the importance of including GHG concentrations inside the RCM domain. Given its significance as clearly shown by this study, this is a novel and important contribution that will be of great interest within the RCM community.

In my opinion, the article is well written and conveys the message very clearly. The experimental setup and the analysis, including the statistical tests used to quantify statistical significance, are straightforward and rigorous leading to little room for ambiguity in the results.

Most details about the experiments and the analysis are provided in the "Methods" section although I noticed that there might be some missing information if we want the experiments to be fully reproducible. For example, I noticed that there is no information about the number of vertical levels and the model time step. Maybe an alternative is to provide the full WRF namelist in the Supplementary Material.

In agreement with my comments above, I totally support this article for publication. I have however two comments that might help to make the paper even stronger.

First, while I agree about the lack of documentation on this issue in most papers and more generally in the specifications of intercomparison projects (e.g., CORDEX), it is unclear to me to what extent this is a real issue in practice. The authors mention "...we found out that some of them does actually include varying GHG concentrations in their RCM runs; while others do not." I think it would be very useful to have a more precise diagnostic on the proportion of modelling groups including or not GHG concentrations changes in their runs. This does not need to include all intercomparison projects. A personal account of for example the Euro-CORDEX groups would already provide a good sense about the extent of the problem.

A second point that I think can help to make the paper stronger is to provide more discussion about the general validity of the results given the limited number of experiments used in this study. This should be done not to undermine the results presented here (which have very clear implications) but with the aim of suggesting possible lines of future research surrounding this issue. For example, my understanding is that the pattern of the difference between variable and constant GHG RCM runs should be similar to the pattern of the "spatial spin-up" (see for example Laprise et al. (2009) and some references therein). Following this hypothesis, VAR-CTE differences should increase as we move away from the lateral boundaries and they should show a large dependence on the flow speed and the inflow side. A lot of work has been done in the spatial/vertical spin-up issue that might be useful to better understand differences between VAR and CTE.

The results presented here somewhat support the "spatial spin-up" pattern. For example, in Figure 2 the winter (DJF) figure shows such a pattern. However the reanalysis driven results tend to show largest differences close to the border. Any idea why? Also, would you expect similar results when looking at other variables? What about if considering variables far from the surface?

Laprise, R., D. Kornic, M. Rapaic, L. Separovic, M. Leduc, O. Nikiema, A. Di Luca, E. Diaconescu, A. Alexandru, Ph. Lucas-Picher, R. de Elia, D. Caya and S. Biner, 2010: Considerations of domain size and large-scale driving for nested Regional Climate Models: Impact on internal variability and skill at developing small-scale details. In: *Climate Change: Inferences from Paleoclimate and Regional Aspects*, Proceedings of the Milutin Milankovitch 130th Anniversary Symposium. Belgrade, 22-25 September 2009. Springer, Editors: A. Berger, F. Mesinger and Dj. Sijacki, Part 4, 181-199, DOI: 10.1007/978-3-7091-0973-1-14.

Specific comments:

Main text:

Line 40: maybe good to mention here that there is another dynamical downscaling approach given by variable resolution global climate models.

Line 77: you raise a good point here. However, from your results it is unclear to what extent this is the explanation as RCM runs including a variable concentration of GHG also tend to systematically underestimate temperature trends compared with the driving data. Can you comment on this?

Line 87: is the RCP8.5 very different from the other pathways when looking at the beginning of the 21st century? I thought that until 2050 they were pretty similar and differences were important later. Can you briefly comment on this?

Line 105: "usually depict" as some trends are negative.

Line 113: maybe use "weaker" instead of "smoothed"?

Line 118: "Differences between CTE and VAR runs are larger..." This is generally the case but depends on the region right?

Line 129: I am not sure which land-atmosphere feedback are you referring here. Can you be more precise?

Methods:

Lines 194-195: is there an annual cycle in the GHG concentrations? Is there a linear trend between years?

Line 210: GHG concentration yearly variability is given by the RCP8.5 values. What about for the historical period? Where are these values coming from?

Data analysis:

Lines 223: can you mention here the significance level used in the text? I saw the p-value in some caption but it might be good to mention here too.

Reviewer #3 (Remarks to the Author):

The manuscript "Towards good practice in regional climate modeling: constant vs. evolving GHG concentrations and its impact on the warming signal" demonstrates that including time varying GHG concentrations affects the model results and trend analysis, where GHG trajectories are not defined in the CORDEX experimental design. The future warming found in varying GHG simulations (relative to fixed 2005 concentrations) seems intuitive, but the manuscript stresses this point while not addressing the larger context. The authors note, via personal communication, that some modeling groups participating in EURO-CORDEX do not include time varying GHGs but it is unclear to the reader if this is one or two models or a systematic problem throughout CORDEX. There are not many regional models used in CORDEX, so it would seem reasonable to list which models used fixed GHG vs time varying. The text does not describe why these modeling centers use constant GHGs. Is this choice attributed to using older models, models with limited radiative physics or a general assumption that the GHG radiative forcing within the RCMs are small?

The manuscript text ignores that the influence of RCM GHG warming is very likely dependent upon domain size. The EURO-CORDEX domain is rather large, with the GCM boundary conditions that include transient GHG are forced at the edges. A small domain would likely be insensitive to RCM GHGs. Although it is not the focus of this paper, the text does not discuss that WRF modifies the original GCM anomaly (both CTE and VAR simulations), which is a nontrivial issue for dynamical downscaling. Figure 2 (c & o) indicates the WRF modification of the GCM anomaly is as large or larger than the choice of constant or time varying GHGs. Also see SFig 1-9e for ERA trend modification. The authors successfully make the case that including time varying GHG concentrations do alter the magnitude of future warming and that documenting the evolution of GHG concentrations should be included in the CORDEX experimental design. However, without addressing the comments above to provide context, the text as written is not appropriate as a standalone Nature Communications article.

Minor comments:

The text would also benefit from an English language copyediting service.

Page 1, line 16: "may be not being" perhaps it should be "may not be"?

Page 3, line 79. "that could eventually compromise most of the past research" It has not been established that many or even most RCMs use fixed GHGs.

Page 3, line 86. It is not clear what CTE stands for.

Page 3, line 90. The authors may want to note that over 2006-2050, RCP8.5 and RCP4.5 are very similar. The two scenarios largely diverge in the second half of the century.

Page 3, line 96. It would be useful to note in the main text that the constant experiment CTE uses 2005 GHG concentrations.

Page 4, line 128. It should be made clear the SSTs are constant in the WRF simulations, not the MPI-ESM-LR simulations.

Reviewer #1 (Remarks to the Author):

Authors' response:

We thank the reviewer for devoting time to revise our manuscript and provide constructive feedback. We appreciate the effort and have done our best to address every point.

The authors of the paper in review highlight the poor level of documentation in the current literature of the capability of Regional Climate Models (RCM) to use same climate global forcing data as used by the boundary forcing GCM.

In the framework of the CMIP5 experiment, the climate forcing data list has been agreed upon and the list of recommended and required datasets can be found on the project website (<http://cmip-pcmdi.llnl.gov/cmip5/forcing.html>). The experiment design document (Taylor 2009/2011) describes for the Historical experiment to include:

- atmospheric composition (including CO₂), due to both anthropogenic and volcanic influences
- solar forcing
- emissions or concentrations of short-lived species and natural and anthropogenic aerosols or their precursors
- land use

while for each of the RCP experiment the prescribed GHG concentration changes were required to facilitate direct comparison between models with and without a carbon cycle component.

For the CORDEX experiment, Gutowsky et al paper of 2016 (WCRP COordinated Regional Downscaling EXperiment WCRP COordinated Regional Downscaling EXperiment (CORDEX): a diagnostic MIP for CMIP6) outlines the importance of GHG concentration as a forcing also for the RCM physical downscaling, together with the land-use and aerosol forcing effects. In Jones 2011 (Jones C, F. Giorgi and G. Asrar, 2011: The Coordinated Regional Downscaling Experiment: CORDEX, An international downscaling link to CMIP5: CLIVAR Exchanges, No. 56, Vol 16, No.2 pages 34-40) it is written:

"The main sources of uncertainty in regional climate change projections can be summarized as: (i) GHG emission uncertainty, (ii) CGCM differences, (iii) CGCM internal variability, (iv) RCD method/model/resolution differences and (v) RCD internal variability. Furthermore, it is also possible that systematic errors in either or both CGCMs and RCMs can affect the realism of simulated climate projections. The relative contribution of all these sources to the total uncertainty will be dependent on the region of interest."

Thus in the CORDEX experiment framework the importance of the GHG forcing also at regional scale is emphasized.

Nevertheless, in net contrast with the high level of detail provided for CMIP5 experiment framework, the CORDEX experiment framework provides scant information on the RCM model capabilities to use the same climate forcing data used by the parent GCMs, and concentrate mostly in describing regional domains and nesting GCM. On the CORDEX project website, in the experiment protocol there is no explicit mention to the climate forcing capabilities of the RCM, and in the data format and archive design no information is required on climate forcing. Only information required is the detail of the boundary GCM forcing.

The revised version of the paper includes references to the works of Gutowsky et al. (2016) and Jones et al. (2011) and mentions the considerations made by the reviewer (see lines 68-91). This way we have further emphasized that while the importance of the GHG forcing for the RCM downscaling has been pointed out in these works, it is implicitly obviated in the CORDEX experiment framework in particular (in contrast with the high level of detail provided for the CMIP5 experiment) as well as in most of the literature based on RCM experiments.

The authors of the paper in review make two strong claims:

1) on line 51-55 : "Some radiation schemes, as implemented in RCMs, prescribe by default GHG concentrations to fixed values, assuming that the GHG atmospheric composition change has little effects on the regional climate with the effect of global climate change being sufficiently included through the GCMs-provided boundary conditions."

2) on line 58-60: "In personal communication with various modeling groups involved in the EURO-CORDEX initiative, we found out that some of them include varying GHG concentrations in their RCM runs; while others do not"

The first claim is both in contrast with lines 50-51 of the paper in review "radiative schemes [...] are heavily affected by the concentration of GHG on the atmosphere", the CORDEX experiment aim as described in the Jones and Gutowski papers, and the evidence at large of the climate change as having an anthropogenic source from the GHG concentration increase. This claim is moreover shown to be false by the paper itself. Where this assumption comes from? To my knowledge, no research paper making this claim has shown evidence of it to the scientific community.

The first claim was simply based on the fact that "Some radiation schemes, as implemented in RCMs, prescribe by default GHG concentrations to fixed values". It is not that we agree with or have evidences of that - not at all. Indeed, as the reviewer says, it is in contrast with our results. For the sake of clarity, we have rewritten the relevant paragraph, see lines 55-67.

The second claim, that some EURO-CORDEX model do not implement varying GHG as prescribed in the RCP used by the parent driving GCM, it must be detailed and a list of this models should be provided. The statement that "[...] we intend to present no judgment on the methodology followed by regional modeling groups worldwide [...]" is not enough on scientific point of view: science should not be interested in not hurting prides, but in spotting errors and correct them to better describe reality.

The authors should at least provide for the EURO-CORDEX experiment they claim to have information about a table of the models with flag if they implement or not varying GHG.

The manuscript provides now more detail on which models runs within Euro-Cordex use evolving GHG concentrations and which do not (to date, to the best of our knowledge, see lines 72-75).

We have changed the previous 'politically correct' statement to a scientifically correct one. See lines 241-243.

The paper provides a good experiment to outline the importance of using the prescribed GHG concentration data from the RCP in the RCM consistent with the one used by the driving GCM.

On the data analysis, the regional impact of the varying GHG is well detailed and consistent with one would expect but the attribution to model internal variability should be excluded also for completeness by performing a small perturbation experiment of the CTE run.

In response to the comments by the Editor and reviewers #2 and #3, we have enlarged our experimental set by performing an ensemble of regional simulations to see the consistency and coherence of our findings beyond a particular experimental setup. See Figure 3 and the last subsection in the Results section. We think that this additional analysis addresses also the latter point raised by the reviewer by filtering out residuals from model internal variability (see for instance the ensemble mean patterns). Besides, the impact of the model internal variability on the trend estimate is known to be weak from the existing literature (e.g., Nabat et al., 2014).

On the issue of reproducibility, a stronger and more effective request to the community of Regional Climate modelers would be to include any relevant parameter required to run the scenario simulation contributed to the ESGF repository as metadata in the model output file itself. In the authors paper itself, some 10 lines were required to describe the RCM model setup,

but for other RCM models where any physical parametrization is tunable, not providing the full list of physically relevant parameters most often prevent any other research group from replicating the experiment and thus falsification may be impossible without direct interaction with the data provider.

We now provide more detail on the model setup and the WRF namelists as used in our experimental setup as Supplementary Material. This guarantees full reproducibility of our experiments.

References:

Nabat, P., Somot, S., Mallet, M., Sanchez-Lorenzo, A., & Wild, M. (2014). Contribution of anthropogenic sulfate aerosols to the changing Euro-Mediterranean climate since 1980. *Geophysical Research Letters*, 41(15), 5605-5611.

Reviewer #2 (Remarks to the Author):

Authors' response:

We thank the reviewer for devoting time to revise our manuscript and provide constructive feedback. We appreciate the effort and have done our best to address every point.

Review of "Towards good practice in regional climate modeling: constant vs. evolving GHG concentrations and its impact on the warming signal" by S. Jerez, J. M. López-Romero, M. Turco, P. Jiménez-Guerrero, R. Vautard and J. P. Montávez

General comments:

The main aim of this paper is to call attention about an often undocumented and certainly understudied practice regarding the inclusion of greenhouse gases (GHG) concentrations as an evolving external forcing within climate simulations using limited-area nested Regional Climate Models. A second aim is to assess the effect of constant Vs. varying GHG by looking at 2-m temperature trends in twin simulations (with/without varying GHG) performed over Europe using a single RCM driven by two different global datasets (a GCM and a reanalysis). Their results show that temperature recent past trends and near future changes are significantly different when including the effect of GHG within the RCM domain. The general effect is shown to be important when considering annual mean temperatures but also seasonal (JJA and DJF) mean minimum and maximum temperatures.

Since the inception of RCMs nearly 25 years ago, the RCM community has devoted important efforts to assess the sensitivity of a number of experimental choices within the RCM setup including the domain size and position, spatial resolution jump between the RCM and the driving data, nudging technique at the boundaries and in the interior, and others. However, as the authors note in the text, (and as far as I know) this is the first study quantifying the importance of including GHG concentrations inside the RCM domain. Given its significance as clearly shown by this study, this is a novel and important contribution that will be of great interest within the RCM community.

In my opinion, the article is well written and conveys the message very clearly. The experimental setup and the analysis, including the statistical tests used to quantify statistical significance, are straightforward and rigorous leading to little room for ambiguity in the results. Most details about the experiments and the analysis are provided in the "Methods" section although I noticed that there might be some missing information if we want the experiments to be fully reproducible. For example, I noticed that there is no information about the number of vertical levels and the model time step. Maybe an alternative is to provide the full WRF namelist in the Supplementary Material.

We now provide more detail on the model setup and the WRF namelists as used in our experimental setup as Supplementary Material. This guarantees full reproducibility of our experiments.

In agreement with my comments above, I totally support this article for publication. I have however two comments that might help to make the paper even stronger. First, while I agree about the lack of documentation on this issue in most papers and more generally in the specifications of intercomparison projects (e.g., CORDEX), it is unclear to me to what extent this is a real issue in practice. The authors mention "...we found out that some of them does actually include varying GHG concentrations in their RCM runs; while others do not." I think it would be very useful to have a more precise diagnostic on the proportion of modelling groups including or not GHG concentrations changes in their runs. This does not need to include all intercomparison projects. A personal account of for example the Euro-CORDEX groups would already provide a good sense about the extent of the problem.

The manuscript provides now more detail on which models runs within Euro-Cordex use evolving GHG concentrations and which do not (to date, to the best of our knowledge, see lines 72-75). Note, however, that we avoid the provision of a detailed list of Euro-Cordex simulations for two main reasons: (1) such a list would be constantly changing (with new simulations being added and others removed or

updated) since Euro-Cordex is a live initiative, and (2) the objective of this contribution is to call the attention on an issue that effectively affects a benchmark coordination initiative such as Euro-Cordex, but whose effective extent might transcend it.

A second point that I think can help to make the paper stronger is to provide more discussion about the general validity of the results given the limited number of experiments used in this study. This should be done not to undermine the results presented here (which have very clear implications) but with the aim of suggesting possible lines of future research surrounding this issue.

We agree that it was necessary to provide evidence of the consistency and extent of our results in a wider context, beyond a particular simulation, in order to broaden their appeal and reinforce our message. After a deep discussion on how best to proceed, we proposed running a multi-physics ensemble of simulations to the Editor. The proposal was approved by the Editor conditioned to your opinion. It was based on several reasons. One one hand, it has been widely proven in the literature that different physical configurations of regional models provide the same range of uncertainty (ensemble spread) as an ensemble of different models, as they may, in fact, rely on different physical configurations (e.g. Jerez et al. 2013a,b; García-Díez et al., 2015). On the other, physical schemes or parameterizations are common across models. That is, the same schemes that are available for our regional climate model are also included in other models. Hence, the results from the proposed multi-physics ensemble shall be actually straightforward relevant for the wide regional climate modeling community. Last, this approach also presented the advantage of being demanding but feasible, allowing us to be self-sufficient in addressing the very time-consuming task of performing an ensemble of climate simulations. This made it possible to have the revised paper ready in a few months, which is also a plus given the urgency of warning the scientific community (from modelers to end-users of climate information) about our findings.

So we designed and ran an ensemble of WRF simulations that include modifications in the radiative scheme (radiative schemes handle the external forcing giving by the GHG atmospheric concentrations), the microphysics and the convective schemes, totalizing a number of 4 ensemble members (see Methods). The results are presented in Figure 3 and the last subsection in the Results section.

For example, my understanding is that the pattern of the difference between variable and constant GHG RCM runs should be similar to the pattern of the “spatial spin-up” (see for example Laprise et al. (2009) and some references therein). Following this hypothesis, VAR-CTE differences should increase as we move away from the lateral boundaries and they should show a large dependence on the flow speed and the inflow side. A lot of work has been done in the spatial/vertical spin-up issue that might be useful to better understand differences between VAR and CTE.

The results presented here somewhat support the “spatial spin-up” pattern. For example, in Figure 2 the winter (DJF) figure shows such a pattern. However the reanalysis driven results tend to show largest differences close to the border. Any idea why? Also, would you expect similar results when looking at other variables? What about if considering variables far from the surface?

Laprise, R., D. Kornic, M. Rapaic, L. Separovic, M. Leduc, O. Nikiema, A. Di Luca, E. Diaconescu, A. Alexandru, Ph. Lucas-Picher, R. de Elia, D. Caya and S. Biner, 2010: Considerations of domain size and large-scale driving for nested Regional Climate Models: Impact on internal variability and skill at developing small-scale details. In: *Climate Change: Inferences from Paleoclimate and Regional Aspects*, Proceedings of the Milutin Milankovitch 130th Anniversary Symposium. Belgrade, 22-25 September 2009. Springer, Editors: A. Berger, F. Mesinger and Dj. Sijacki, Part 4, 181-199, DOI: 10.1007/978-3-7091-0973-1-14.

In order to address the reviewer's wise comment on the effect of the domain size, a new set of simulations (with and without varying GHG concentrations) has been conducted, identical to the original ones but over a smaller area domain covering part of central Europe, where we found some of the largest signals (see Supp. Fig. 13 and the comment in lines 192-194). Also, the general "spatial spin-up" issue has now been introduced in the second paragraph of the main paper based on the work by Laprise et al. (2010) and others.

Regarding how TAS trends are reproduced when WRF is driven by ERA, note that differences between the VAR and CTE runs appear over the areas where both VAR and CTE runs underestimate TAS trends the most, as compared to ERA (Fig. 1, first row). This (1) links with a comment below: "the inclusion of variable GHG in RCMs smooths the systematic underestimation of temperature trends as compared to the driving global datasets, although such underestimation, even if weaker, persists", so that's why VAR-CTE differences appear close to the borders, and (2) moves the reviewer's question one step before: why does the largest underestimation of TAS trends (as reproduced by the RCM runs, when compared to ERA) appear close to the domain borders (where boundary conditions are supposed to exert their maximum control though)? Response to this question is out of the scope of this contribution, but may be framed on how RCMs solve clouds, handle aerosols, model land-atmosphere feedbacks and treat SST (see fourth paragraph of the main paper).

In the vertical direction, we can confirm that signals are attenuated with height (see Figure 1 below). However, we prefer not to include a discussion about this fact in the main paper on behalf of its brevity and to avoid deviations from its main message (behaviors in the upper troposphere and the stratosphere may be the focus of a separate contribution). For the same reason, we prefer to focus only on near surface air temperature (the variable whose link to the GHG forcing is most evident and straightforward); others variables might present more or less similar sensitivity patterns depending on their more or less linear relationship with the temperature, in particular, and with the GHG forcing, in general.

Figure 1: CTE (first column) and VAR (second column) projections (2011-2040 minus 1971-2000) for annual TAS (surface air temperature, first row) and T850 (temperature at 850 hPa, second row) along with the VAR minus CTE differences in such projections (third column). Points in pannel c indicate that the magnitude of the difference between VAR and CTE experiments equals or is greater than the magnitude of the signal from the respective CTE experiment. Only significant values ($p < 0.1$) are shown. Units: K.

Specific comments:

Main text:

Line 40: maybe good to mention here that there is another dynamical downscaling approach given by variable resolution global climate models.

Done.

Line 77: you raise a good point here. However, from your results it is unclear to what extent this is the explanation as RCM runs including a variable concentration of GHG also tend to systematically underestimate temperature trends compared with the driving data. Can you comment on this?

The inclusion of variable GHG in RCMs, based on our results, smooths the systematic underestimation of temperature trends as compared to the driving GCMs, although such underestimation, even if weaker, persists. Other causes should thus be looked for among those already mentioned in the manuscript based on the existing literature: presumably, inadequacies in the characterization and modeling of cloud formation and soil properties (see fourth paragraph).

Line 87: is the RCP8.5 very different from the other pathways when looking at the beginning of the 21st century? I thought that until 2050 they were pretty similar and differences were important later. Can you briefly comment on this?

Done. See lines 110-115.

Line 105: "usually depict" as some trends are negative.

Done. See line 131.

Line 113: maybe use "weaker" instead of "smoothed"?

Done. See line 140.

Line 118: "Differences between CTE and VAR runs are larger..." This is generally the case but depends on the region right?

Nuanced. See lines 147-149.

Line 129: I am not sure which land-atmosphere feedback are you referring here. Can you be more precise?

Clarified. See lines 158-160.

Methods:

Lines 194-195: is there an annual cycle in the GHG concentrations? Is there a linear trend between years?

Line 210: GHG concentration yearly variability is given by the RCP8.5 values. What about for the historical period? Where are these values coming from?

In our experiments, the assimilated values of GHG atmospheric concentrations are annual estimations (based on inventory data for the historical period and on socio-economic projections for the scenario period), as provided by the RCP data base (<http://tntcat.iiasa.ac.at:8787/RcpDb>), showing a positive trend during the last decades that is projected to continue in the next ones (e.g. IPCC, 2013). This is now better clarified in the text (see Methods, lines 261-268).

Data analysis:

Lines 223: can you mention here the significance level used in the text? I saw the p-value in

some caption but it might be good to mention here too.

Done. See lines 316 and 333.

References:

García-Díez, M., Fernández, J., & Vautard, R. (2015). An RCM multi-physics ensemble over Europe: multi-variable evaluation to avoid error compensation. *Climate dynamics*, 45(11-12), 3141-3156.

IPCC (2013). *Climate Change 2013: The Physical Science Basis. Contribution of Working Group I to the Fifth Assessment Report of the Intergovernmental Panel on Climate Change* [Stocker, T.F., D. Qin, G.-K. Plattner, M. Tignor, S.K. Allen, J. Boschung, A. Nauels, Y. Xia, V. Bex and P.M. Midgley (eds.)]. Cambridge University Press, Cambridge, United Kingdom and New York, NY, USA, 1535 pp.

Jerez, S., Montavez, J. P., Gomez-Navarro, J. J., Lorente-Plazas, R., Garcia-Valero, J. A., & Jimenez-Guerrero, P. (2013a). A multi-physics ensemble of regional climate change projections over the Iberian Peninsula. *Climate dynamics*, 41(7-8), 1749-1768.

Jerez, S., Montavez, J. P., Jimenez-Guerrero, P., Gomez-Navarro, J. J., Lorente-Plazas, R., & Zorita, E. (2013b). A multi-physics ensemble of present-day climate regional simulations over the Iberian Peninsula. *Climate dynamics*, 40(11-12), 3023-3046.

Reviewer #3 (Remarks to the Author):

Authors' response:

We thank the reviewer for devoting time to revise our manuscript and provide constructive feedback. We appreciate the effort and have done our best to address every point.

The manuscript "Towards good practice in regional climate modeling: constant vs. evolving GHG concentrations and its impact on the warming signal" demonstrates that including time varying GHG concentrations affects the model results and trend analysis, where GHG trajectories are not defined in the CORDEX experimental design. The future warming found in varying GHG simulations (relative to fixed 2005 concentrations) seems intuitive, but the manuscript stresses this point while not addressing the larger context.

We agree that it was necessary to provide evidence of the consistency and extent of our results in a wider context, beyond a particular simulation, in order to broaden their appeal and reinforce our message. After a deep discussion on how best to proceed, we proposed running a multi-physics ensemble of simulations to the Editor. The proposal was approved by the Editor conditioned to your opinion. It was based on several reasons. One one hand, it has been widely proven in the literature that different physical configurations of regional models provide the same range of uncertainty (ensemble spread) as an ensemble of different models, as they may, in fact, rely on different physical configurations (e.g. Jerez et al. 2013a,b; García-Díez et al., 2015). On the other, physical schemes or parameterizations are common across models. That is, the same schemes that are available for our regional climate model are also included in other models. Hence, the results from the proposed multi-physics ensemble shall be actually straightforward relevant for the wide regional climate modeling community. Last, this approach also presented the advantage of being demanding but feasible, allowing us to be self-sufficient in addressing the very time-consuming task of performing an ensemble of climate simulations. This made it possible to have the revised paper ready in a few months, which is also a plus given the urgency of warning the scientific community (from modelers to end-users of climate information) about our findings.

So we designed and ran an ensemble of WRF simulations that include modifications in the radiative scheme (radiative schemes handle the external forcing giving by the GHG atmospheric concentrations), the microphysics and the convective schemes, totalizing a number of 4 ensemble members (see Methods). The results are presented in Figure 3 and the last subsection in the Results section.

The authors note, via personal communication, that some modeling groups participating in EURO-CORDEX do not include time varying GHGs but it is unclear to the reader if this is one or two models or a systematic problem throughout CORDEX. There are not many regional models used in CORDEX, so it would seem reasonable to list which models used fixed GHG vs time varying. The text does not describe why these modeling centers use constant GHGs. Is this choice attributed to using older models, models with limited radiative physics or a general assumption that the GHG radiative forcing within the RCMs are small?

The manuscript provides now more detail on which models runs within Euro-Cordex use evolving GHG concentrations and which do not (to date, to the best of our knowledge, see lines 72-75). We are, however, not able to speculate the reasons why those models are being used with constant GHG concentrations for long-term runs. They are simply wrong. The only reason we can think is, in the case of the WRF model, that its default compilation does not include the varying GHG forcing, as commented in the text (lines 57-60).

The manuscript text ignores that the influence of RCM GHG warming is very likely dependent upon domain size. The EURO-CORDEX domain is rather large, with the GCM boundary conditions that include transient GHG are forced at the edges. A small domain would likely be insensitive to RCM GHGs. Although it is not the focus of this paper, the text does not discuss

that WRF modifies the original GCM anomaly (both CTE and VAR simulations), which is a nontrivial issue for dynamical downscaling. Figure 2 (c & o) indicates the WRF modification of the GCM anomaly is as large or larger than the choice of constant or time varying GHGs. Also see SFig 1-9e for ERA trend modification. The authors successfully make the case that including time varying GHG concentrations do alter the magnitude of future warming and that documenting the evolution of GHG concentrations should be included in the CORDEX experimental design. However, without addressing the comments above to provide context, the text as written is not appropriate as a standalone Nature Communications article.

In order to address the reviewer's wise comment on the effect of the domain size, a new set of simulations (with and without varying GHG concentrations) has been conducted, identical to the original ones but over a smaller area domain covering part of central Europe, where we found some of the largest signals (see Supp. Fig. 13 and the comment in lines 192-194). Also, the general "spatial spin-up" issue has now been introduced in the second paragraph of the main paper based on the work by Laprise et al. (2010) and others.

On the other hand, following the reviewer's indication, we now further emphasize the discrepancies between the WRF simulations and the driving global datasets and mention explicitly that the differences between VAR and CTE are generally smaller than between each WRF realization and the global datasets in most cases (see lines 144-147 and 186-190).

Minor comments:

The text would also benefit from an English language copyediting service.

We have had the paper proofread by a native English speaker.

Page 1, line 16: "may be not being" perhaps it should be "may not be"?

Changed. See line 16.

Page 3, line 79. "that could eventually compromise most of the past research" It has not been established that many or even most RCMs use fixed GHGs.

Changed. See lines 102-103.

Page 3, line 86. It is not clear what CTE stands for.

Clarified. See lines 106-110.

Page 3, line 90. The authors may want to note that over 2006-2050, RCP8.5 and RCP4.5 are very similar. The two scenarios largely diverge in the second half of the century.

Done. See lines 110-115.

Page 3, line 96. It would be useful to note in the main text that the constant experiment CTE uses 2005 GHG concentrations.

Done. See lines 109-110.

Page 4, line 128. It should be made clear the SSTs are constant in the WRF simulations, not the MPI-ESM-LR simulations.

Actually, we meant that SST was constant (=identical) between the CTE and VAR experiments, but it does evolve also in the regional experiments. Better clarified now in lines 157-158.

References:

García-Díez, M., Fernández, J., & Vautard, R. (2015). An RCM multi-physics ensemble over Europe: multi-variable evaluation to avoid error compensation. *Climate dynamics*, 45(11-12), 3141-3156.

Jerez, S., Montavez, J. P., Gomez-Navarro, J. J., Lorente-Plazas, R., Garcia-Valero, J. A., & Jimenez-Guerrero, P. (2013a). A multi-physics ensemble of regional climate change projections over the Iberian Peninsula. *Climate dynamics*, 41(7-8), 1749-1768.

Jerez, S., Montavez, J. P., Jimenez-Guerrero, P., Gomez-Navarro, J. J., Lorente-Plazas, R., & Zorita, E. (2013b). A multi-physics ensemble of present-day climate regional simulations over the Iberian Peninsula. *Climate dynamics*, 40(11-12), 3023-3046.

REVIEWERS' COMMENTS:

Reviewer #1 (Remarks to the Author):

The authors have successfully addressed the reviewer comments. The paper in its actual form has both scientific and formal content suitable for the publication.

The spotlight on varying GHG gases concentrations used in WRF RCM simulations is well presented and the recommendation to create a better defined framework for RCM down-scaling experiment contained in this paper can provide a basis for new experiment guidelines.

Reviewer #2 (Remarks to the Author):

Second review of "Towards good practice in regional climate modeling: constant vs. evolving GHG concentrations and its impact on the warming signal" by S. Jerez, J. M. López-Romero, M. Turco, P. Jiménez-Guerrero, R. Vautard and J. P. Montávez

General comments:

The authors have successfully addressed the major comments I have made in the first review. As mentioned in my first review, I believe the issue presented in this paper is novel and of great interest for the regional climate modelling community but also for those using RCM derived data. The relevance of the use of constant or variable greenhouse gases concentrations within the RCM domain is well shown. As expected, the paper does not answer all questions surrounding the issue and thus it might motivate some more new research.

In my opinion the article is now ready for publication. I have however a few minor comments that the authors might consider.

1. One is about the use of 1.5 K threshold instead of using of a fixed period as usually done. In my opinion using a temperature threshold leads to some confusion without having any advantage. For example, the following sentence "The results show that the TAS trends are significantly affected by 1-2 K century⁻¹, which under 1.5oC global warming translates into a non-negligible impact of up to 1 K in the future projections of TAS, similarly affecting projections for maximum and minimum temperatures" would be clearer if using some fix present and future periods.

2. Also, the use of a temperature threshold has the disadvantage that the corresponding future period will be dependent on the simulation considered. From the text it seems that the authors take the same period for all simulations (30-year period centred in 2025) so it is unclear to me whether they use a temperature threshold or a fix period. Again, I don't see the advantage of the 1.5 K threshold.

3. A second general comment is about the number of panels in each figure. In my opinion the number of plots in some figures could be reduced so that the message you want to convey is more emphasised. Is it necessary to include seasonal differences in Figures 2 and 3? Is it necessary to include future (third row) results in Figure 1? Is it necessary to include all the differences in all figures? Is it necessary to show four different metrics to quantify the sensitivity in Figure 3? There is a lot of information in the Supplementary Material that is relevant but secondary and I think that some more information could be moved there. The key is to think what is the message of each figure.

4. Also, for the reader it would be nice to include the variable with units next to the colourbar, not only in the caption. It is a little annoying to go back and forth between the figure and the caption to understand what is plotted.

5. Why not denoting the default simulation as CONTROL instead of Exp. 2?

6. Finally, I think it would be nice to include some possible future lines of research in the conclusions. For example, do you think it would be useful to evaluate this issue with another RCM? Do you think that other variables might be affected differently than TAS?

Specific comments:

Line 19: remove "growing"

Line 156: remove "the highly inertial"

Line 201: "It should be noted that different physical configurations of regional models provide the same range of uncertainty (or ensemble spread) as an ensemble of different models, as they may indeed rely on different physical configurations (Jerez et al. 2013a,b; García-Díez et al., 2015)." This is partially true as it depends on the variable considered. For example, Di Luca et al (2014) found that the multi-physics spread is as large as a multi-model spread when considering precipitation but smaller when considering evaporation, a variable that is less dependent on the physics. I would assume that TAS would behave more similar to evaporation than to precipitation. Figure 1 caption: this caption reads "Simulated trends of TAS from global simulations and regional experiments" but it includes reanalysis data. Can you modified this so reanalysis are included?

Di Luca, A., E. Flaounas, P. Drobinski and C. Lebeaupin-Brossier, 2014 : The atmospheric component of the Mediterranean Sea water budget in a WRF multi-physics ensemble and observations. *Climate Dynamics*, 43 :2349-2375.

Reviewer #3 (Remarks to the Author):

The authors have addressed all of my comments and concerns in their revision. I believe the authors have also addressed the comments from the other two reviewers, but I did not evaluate their comments thoroughly.

I am not sure the MSE column is needed in Figure 3 (b, f and j). Identifying which scheme is the most sensitive to GHGs seems a bit outside of the scope of the paper. It is not obvious that the numbers in the MSE column correspond to the experiment numbers in Table 1. Likewise, the symbols in the third and fourth columns are so small I am not sure they are needed.

Reviewer #1 (Remarks to the Author):

Authors' response:

The authors have successfully addressed the reviewer comments. The paper in its actual form has both scientific and formal content suitable for the publication.

The spotlight on varying GHG gases concentrations used in WRF RCM simulations is well presented and the recommendation to create a better defined framework for RCM down scaling experiment contained in this paper can provide a basis for new experiment guidelines.

We do thank the reviewer for the time devoted to revise our manuscript.

Reviewer #2 (Remarks to the Author):

Authors' response:

Second review of "Towards good practice in regional climate modeling: constant vs. evolving GHG concentrations and its impact on the warming signal" by S. Jerez, J. M. López-Romero, M. Turco, P. Jiménez-Guerrero, R. Vautard and J. P. Montávez

General comments:

The authors have successfully addressed the major comments I have made in the first review. As mentioned in my first review, I believe the issue presented in this paper is novel and of great interest for the regional climate modelling community but also for those using RCM derived data. The relevance of the use of constant or variable greenhouse gases concentrations within the RCM domain is well shown. As expected, the paper does not answer all questions surrounding the issue and thus it might motivate some more new research.

In my opinion the article is now ready for publication. I have however a few minor comments that the authors might consider.

We do thank the reviewer for the time devoted to revise our manuscript.

1. One is about the use of 1.5 K threshold instead of using of a fixed period as usually done. In my opinion using a temperature threshold leads to some confusion without having any advantage. For example, the following sentence "The results show that the TAS trends are significantly affected by 1-2 K century⁻¹, which under 1.5°C global warming translates into a non-negligible impact of up to 1 K in the future projections of TAS, similarly affecting projections for maximum and minimum temperatures" would be clearer if using some fix present and future periods.

We fix the future period (2011-2040) as the 30-yr long period for which the global mean surface air temperature, in the WRF-driving GCM simulation, reaches 1.5°C above the pre-industrial level. This approach has the advantage of making our results independent (at least at some degree) of the warming rate provided by a particular GCM and scenario, as the regional climate change projections are made for a warming threshold instead of for a fixed future period. This approach is being more and more adopted nowadays and the IPCC is elaborating a report devoted to impacts under 1.5°C global warming (to which our paper will contribute) partly aimed at elucidating impacts in the context of the Paris agreement.

2. Also, the use of a temperature threshold has the disadvantage that the corresponding future period will be dependent on the simulation considered. From the text it seems that the authors take the same period for all simulations (30-year period centred in 2025) so it is unclear to me whether they use a temperature threshold or a fix period. Again, I don't see the advantage of the 1.5 K threshold.

We used a fixed period (2011-2040) because we used only one GCM simulation. In order to evaluate impacts under 1.5°C global warming, the procedure is to look for the period (a 30-yr long period for climatological purposes) when the global mean surface air temperature reaches 1.5°C above the pre-industrial level. This should be done using the GCM simulation. Once we have identified this future period, we evaluate impacts (as future minus present climatologies) in our set of regional

simulations.

3. A second general comment is about the number of panels in each figure. In my opinion the number of plots in some figures could be reduced so that the message you want to convey is more emphasised. Is it necessary to include seasonal differences in Figures 2 and 3? Is it necessary to include future (third row) results in Figure 1? Is it necessary to include all the differences in all figures? Is it necessary to show four different metrics to quantify the sensitivity in Figure 3? There is a lot of information in the Supplementary Material that is relevant but secondary and I think that some more information could be moved there. The key is to think what is the message of each figure.

Our message is that including the evolution of the GHG atmospheric concentrations in the RCM runs matters, and thus it should be done and documented. In order to provide a quantitative context for our message, the seasonal analysis in Figures 2 and 3 shows upper and bottom bounds of the effect of evolving GHG concentrations in RCM runs (within our experimental set and our analysis approach). We found it interesting to show these bounds and the asymmetry of the impacts depending on the season. Also, although it is clear that the most important column in Figures 1 and 2 is the last one, we found also interesting to show that, while both WRF runs resemble the overall structure of the driving dataset (columns 1, 2 and 4), the patterns of differences with the driving datasets have significant values and differs between the VAR and the CTE experiments, with amplified or smoothed signals, and even with sign-changed signals (columns 3 and 5). This provides a complete context for the results shown in the last column (VAR-CTE differences). Last, each metric in Figure 3 teaches something different, and each one complements each other. So we decided to maintain the Figures as they were. Also because Nature Communications, to aid the reader, prefers to include as much of the Supplementary Information in the main article as possible.

4. Also, for the reader it would be nice to include the variable with units next to the colourbar, not only in the caption. It is a little annoying to go back and forth between the figure and the caption to understand what is plotted.

Done.

5. Why not denoting the default simulation as CONTROL instead of Exp. 2?

Done.

6. Finally, I think it would be nice to include some possible future lines of research in the conclusions. For example, do you think it would be useful to evaluate this issue with another RCM? Do you think that other variables might be affected differently than TAS?

Done.

Specific comments:

Line 19: remove "growing"

Changed: growing → rising

Line 156: remove "the highly inertial"

Done.

Line 201: "It should be noted that different physical configurations of regional models provide the same range of uncertainty (or ensemble spread) as an ensemble of different models, as they may indeed rely on different physical configurations (Jerez et al. 2013a,b; García-Díez et al., 2015)." This is partially true as it depends on the variable considered. For example, Di Luca et al (2014) found that the multi-physics spread is as large as a multi-model spread when considering precipitation but smaller when considering evaporation, a variable that is less dependent on the physics. I would assume that TAS would behave more similar to evaporation than to precipitation.

The cited papers (Jerez et al. 2013a,b; García-Díez et al., 2015) do prove that the spread in TAS in multi-physics ensembles is as large as in multi-model ensembles. We have added the new reference.

Figure 1 caption: this caption reads "Simulated trends of TAS from global simulations and regional experiments" but it includes reanalysis data. Can you modified this so reanalysis are included?

Done.

Di Luca, A., E. Flaounas, P. Drobinski and C. Lebeaupin-Brossier, 2014 : The atmospheric component of the Mediterranean Sea water budget in a WRF multi-physics ensemble and observations. *Climate Dynamics*, 43 :2349-2375.

Reviewer #3 (Remarks to the Author):

Authors' response:

The authors have addressed all of my comments and concerns in their revision. I believe the authors have also addressed the comments from the other two reviewers, but I did not evaluate their comments thoroughly.

We do thank the reviewer for the time devoted to revise our manuscript.

I am not sure the MSE column is needed in Figure 3 (b, f and j). Identifying which scheme is the most sensitive to GHGs seems a bit outside of the scope of the paper. It is not obvious that the numbers in the MSE column correspond to the experiment numbers in Table 1. Likewise, the symbols in the third and fourth columns are so small I am not sure they are needed.

The reviewer is right in the sense that identifying which WRF configuration is the most sensitive to the GHG handling is outside of the scope of the paper. Here, by identifying the MSE, our goal is just to show that all the configurations in the ensemble of simulations are certainly sensitive (and to a comparable extent) to the GHG handling. This provides evidence of the consistency and extent of our results.

We have made clear that the MSE numbers in Figure 3 correspond to the experiments identifiers of Table 1, thanks for the warning.

We agree that the symbol type in the last two columns of Figure 3 is hardly identifiable in the A4 printed version of the figure. However, in its full resolution online version, the interested readers could easily appreciate it. We believe that these symbols (what they represent) reinforce the structural character of our findings.